# Lost in Communication: Do Family Physicians Provide Patients with Information on Preventing Diet-Related Diseases?

**DOI:** 10.3390/ijerph191710990

**Published:** 2022-09-02

**Authors:** Robert Olszewski, Justyna Obiała, Karolina Obiała, Jakub Owoc, Małgorzata Mańczak, Klaudia Ćwiklińska, Magdalena Jabłońska, Paweł Zegarow, Jolanta Grygielska, Marzena Jaciubek, Katarzyna Majka, Daria Stelmach, Andrzej Krupienicz, Jacek Rysz, Krzysztof Jeziorski

**Affiliations:** 1Department of Gerontology, Public Health and Didactics, National Institute of Geriatrics, Rheumatology and Rehabilitation, Spartanska 1, 02-637 Warsaw, Poland; 2Department of Ultrasound, Institute of Fundamental Technological Research, Polish Academy of Sciences, Pawinskiego 5B, 02-106 Warsaw, Poland; 3Department of the Prevention of Environmental Hazards and Allergology, Medical University of Warsaw, Banacha 1a, 02-091 Warsaw, Poland; 4Department of Fundamental of Nursing, Medical University of Warsaw, Erazma Ciołka 27, 01-445 Warsaw, Poland; 5Students’ Scientific Group Affiliated to II Department of Obstetrics and Gynecology, Medical University of Warsaw, Żwirki i Wigury 61, 02-091 Warsaw, Poland; 6Department of Nephrology, Hypertension and Family Medicine, Medical University of Lodz, 92-215 Lodz, Poland; 7Maria Sklodowska-Curie National Research Institute of Oncology, W.K. Roentgena 5, 02-781 Warsaw, Poland

**Keywords:** diet, diet-related diseases, communication, prevention, primary care

## Abstract

Diet-related diseases remain leading causes of death in most developed countries around the world. The aim of the study was to compare opinions of patients and family physicians on receiving and providing recommendations about physical activity, diet and use of medication. **Methods:** The questionnaire study was conducted among patients of 36 primary health care clinics in Poland between September 2018 and February 2019. Patients and physicians were interviewed separately by trained researchers. Data from 509 patients and 167 family doctors were analyzed. **Results:** The median age of patients was 44 years (interquartile range: 29–55) and 70% were women. The majority of physicians were women (59%) and the median age was 37 years (IQR: 31–50). There was a significant difference between physicians’ declarations on providing recommendations on diet (92% vs. 39%) and activity (90% vs. 37%) versus patients’ declarations on receiving them. **Conclusions:** The results indicate that there is significant room for improvement in providing patients with proper recommendations on diet and physical activity by their family physicians. Primary care physicians should put more emphasis on clear communication of recommendations on diet and physical activity.

## 1. Introduction

Diet-related diseases such as various types of cancer, cardiovascular diseases (CVD) and diabetes are becoming increasingly prevalent and are leading causes of death in most developed countries around the world [1]. Being overweight or obese often leads to these diseases. They are not only risk factors but are also a heavy burden on societies and economies. On average, life expectancy is reduced by 2.7 years and Gross Domestic Product of the Organisation for Economic Co-operation and Development (OECD) countries is reduced by 3.3% [2].

Preventive actions and education are usually aimed at eliminating or reducing the negative impact of diseases and eventually decreasing mortality. All recommendations for disease prevention from prominent organizations such as the European Society of Cardiology or American Institute for Cancer Research list inappropriate diet and an insufficient level of physical activity as crucial risk factors [3,4,5,6]. Therefore, patient awareness in that area should be considered a key element in reducing these risks. Although there are numerous and diverse ways to achieve this, such as educational campaigns, promoting healthy lifestyle at workplace and sports activities within local communities, the preventive and educational role of family physicians is of particular importance due to their esteem and unique relations with patients. [7]. Nevertheless, there are no standards on providing such recommendations by physicians, and some of them find it uncomfortable to touch upon sensitive topics or declare no adequate skills [8].

The aim of the study was to compare opinions of family physicians and patients on providing recommendations for proper nutrition, physical activity and use of medication.

## 2. Materials and Methods

We used the database of all 844 primary health care clinics from central Poland contracted by the National Health Fund and listed them in random order. Then we contacted clinics and invited them to participate in the study until the number of clinics that agreed to participate reached 12 in each of the three previously selected groups: villages or small towns (≤20,000 residents), medium-sized towns (20,001–500,000 residents) and cities (>500,000). The final response rate was 22.9% with lack of time as the most common reason given for refusal to take part. The total number of 36 clinics was determined by available resources.

Separate, authorial questionnaires for patients and physicians were administered using the Computer Assisted Personal Interview (CAPI) method by trained researchers. We interviewed patients waiting for a visit who agreed to participate. The participation was voluntary and patients were informed about the option to withdraw at any time. We conducted 527 interviews with patients and 170 with physicians, but rejected 18 patients and 3 physicians who withdrew during the process. The final number of participants was 509 patients and 167 physicians.

The ethical approval for this study was obtained from the Bioethics Committee of the National Institute of Geriatrics, Rheumatology and Rehabilitation.

We collected patient data on: socio-demographics (sex, age, education, place of residence) and smoking status. Socio-demographics data were categorized into groups: age (≤60 and >60 years of age), place of residence (village or small town, medium-size town, city).

Information on weight and height of patients was used to calculate Body Mass Index (BMI). We categorized BMI based on the WHO classification: underweight (BMI < 18.5), normal weight (18.5–24.99), overweight (25.0–29.9) and obese (≥30) [9]. We collected data among family physicians on their sex and age.

The statistical analysis was carried out with Statistica 13.0. The normal distribution of continuous variables was verified using the Shapiro–Wilk test. Continuous data were presented as median and interquartile range (IQR), while categorical variables were presented as numbers and percentages.

The Chi-square test was used to assess differences in sociodemographic, lifestyle and health characteristics in groups of patients receiving and not receiving recommendations on physical activity and diet. A *p*-value of *p* < 0.05 was considered statistically significant. This survey was a part of a larger study investigating preventive and educational practices in primary care clinics [10]. Patients were asked for their consent to take part while interviews took typically 12–15 min to conduct.

## 3. Results

Most of the patients’ study sample were: females (70%, *n* = 354), with secondary (43%, *n* = 220) and higher (44.0%, *n* = 224) level of education, living in cities (34%, *n* = 174) and villages or small towns (39%, *n* = 199). The median age of patients was 44 (IQR 29–55). Most of them had never smoked (54%, *n* = 274).

Approximately 90% of family physicians declared they provided information on physical activity and diet to their patients, while only every third patient declared they received such information (*p* < 0.001). This difference was significantly lower in the case of recommendations on medication use (Figure 1).

The rate of providing patients with recommendations on physical activity and diet varied according to age, sex and BMI. Patients significantly more likely to receive information about diet and physical activity were: over 60 years old, men and obese. Older people were more likely to receive recommendations on use of medication (80% vs. 67 %, *p* = 0.020). The level of education was also a factor in receiving recommendations on physical activity and diet: half of the patients with elementary education declared receiving information on physical activity and diet compared to 32–38% of patients with secondary and higher education. However, education was not a factor in receiving information on medication use (*p* = 0.308). We found no significant differences in receiving information on physical activity, diet and medications with regard to the place of residence and smoking (Table 1).

The analysis of data obtained from family physicians showed no significant differences in providing patients with information on physical activity and diet in relation to age (physical activity, *p* = 0.756; diet, *p* = 0.804) or sex (physical activity, *p* = 0.198; diet, *p* = 0.780).

## 4. Discussion

The aim of this study was to compare opinions of physicians and patients about providing recommendations on diet, physical activity and medications. The key finding was that approximately 90% of family physicians said they inform patients about physical activity and diet, however only one in three patients confirmed that (37% and 39% respectively, *p* < 0.001).

It is an important observation in the context of what is already known and recommended on diet-related disease prevention and education. A proper diet and regular physical activity are strongly recommended and considered not only effective, but also cost-efficient [9]. Thus, it is crucial to take adequate action in order to promote prevention based on a diet and physical activity improvements.

The key question arises as to how it is possible that opinions of patients and physicians can differ so significantly. Our finding is to some extent in line with other studies. Piwonska et al. [11] found that only 30% of the general Polish population received any advice on diet or physical activity, while the American study of primary care providers revealed that only 59% of them discussed physical activity with most of their at-risk patients [12]. However, none of the studies confronted opinions of both patients and physicians.

The results indicate that communicating recommendations by family physicians may indeed not be effective, especially in the case of providing information on physical activity and diet. This may suggest that the problem is rooted in poor communication potentially caused by various factors: reluctance or inability of physicians to clearly communicate lifestyle recommendations, the reluctance of patients to accept them, lack of standards or lack of understanding of long-term risks by patients. It should be noted, however, that obese—thus more at risk—patients received recommendations more often than others. Family physicians’ approach to communicate patient health status is related to patient receptiveness, level of readiness and motivation [13]. Orsal et al. [14] found that higher level of health awareness increased patient satisfaction with primary care, thus addressing this phenomenon is in the best interest of physicians and patients at the same time.

When interpreting the results of our study one needs to take into account the local setting. According to the latest OECD data, Poland has the lowest number of practicing physicians per 1000 population in the European Union and one of the highest number of consultations per physician [15], making provision of preventive recommendations less likely than in other countries for simple lack of time. This study is subject to certain limitations. First, self-reported data are prone to constraints such as cognitive or memory bias among participants. Second, there may be a non-response bias as patients voluntarily agreed to participate in the study. Third, patients were surveyed while waiting for their visit, thus their replies referred more to their general experience rather than the actual visit they were about to receive. Our sample also had an overrepresentation of women, which is a well-known phenomenon in medical surveys [16]. However, this does not influence our conclusions and may in fact strengthen them as women in our study received recommendations less frequently than men. Despite its limitations, this study indicates an important area of prevention effectiveness in terms of patients understanding messages they are provided with.

## 5. Conclusions

The results indicate that there is a significant gap between opinions of physicians and patients referring to communication of recommendations on diet and physical activities. Thus, the major conclusion is that patients require more attention from their family physicians in terms of communication, quality or comprehensiveness of information on proper nutrition and physical activity. The finding that these opinions do not differ significantly in case of recommendations on medication use shows that this is possible and could be significantly improved. As our study is subject to some limitations, future studies should investigate in more detail how physicians communicate crucial recommendations in terms of disease prevention, leading perhaps to working out some standards in that area. Prevention and education concerning diet-related diseases should be a meaningful goal for health care systems across the globe, and family physicians—the first contact point and an important source of information for most patients—play a vital role. The question for policymakers is not only whether such information is provided but also how it is provided. Improving physicians’ awareness of quality of their communication is another important point. Although representativeness of our results is limited by local settings, this nevertheless should be the area of interest for other countries.

## Figures and Tables

**Figure 1 ijerph-19-10990-f001:**
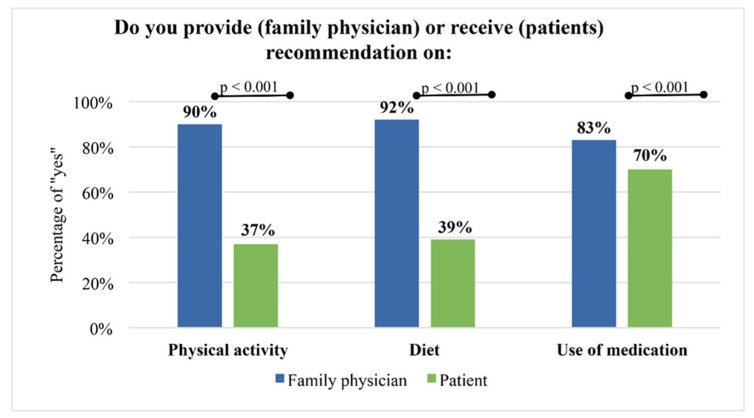
Comparison of the family physicians’ declarations about providing and patients about receiving recommendations on medication use, diet and physical activity.

**Table 1 ijerph-19-10990-t001:** Association of patients characteristics with recommendation on physical activity, diet and use of medication.

		Have You Received Recommendation on:								
	*n* (%)	Physical Activity			Diet			Use of Medication		
		Yes	No	*p*-Value	Yes	No	*p*-Value	Yes	No	*p*-Value
**Age (years)**										
≤60	423 (83)	143 (34)	280 (66)	0.004	148 (35)	275 (65)	<0.001	285 (67)	138 (33)	0.020
>60	86 (17)	43 (50)	43 (50)		49 (57)	37 (43)		69 (80)	17 (20)	
**Sex**										
Female	354 (70)	113 (32)	241 (68)	0.001	127 (36)	227 (64)	0.048	238 (67)	116 (33)	0.086
Male	155 (30)	73 (47)	82 (53)		70 (45)	85 (55)		116 (75)	39 (25)	
**Education**										
Elementary/vocational	65 (13)	34 (52)	31 (48)	0.013	33 (51)	32 (49)	0.083	50 (77)	15 (23)	0.308
Secondary	220 (43)	71 (32)	149 (68)		78 (35)	142 (65)		154 (70)	66 (30)	
Higher	224 (44)	81 (36)	143 (64)		86 (38)	138 (62)		150 (67)	74 (33)	
**Place of residence**										
Village or small town ^a^	199 (39)	63 (32)	136 (68)	0.144	72 (36)	127 (64)	0.517	139 (70)	60 (30)	0.941
Medium-size town ^b^	136 (27)	57 (42)	79 (58)		52 (38)	84 (62)		93 (68)	43 (32)	
City ^c^	174 (34)	66 (38)	108 (62)		73 (42)	101 (58)		122 (70)	52 (30)	
**BMI**										
Underweight ^d^	11 (2)	0 (0)	11 (100)	<0.001	2 (18)	9 (82)	< 0.001	5 (45)	6 (55)	0.099
Normal weight ^e^	263 (52)	83 (32)	180 (68)		82 (31)	181 (69)		175 (67)	88 (33)	
Overweight ^f^	155 (31)	59 (38)	96 (62)		67 (43)	88 (57)		114 (74)	41 (26)	
Obesity ^g^	79 (16)	44 (56)	35 (44)		46 (58)	33 (42)		59 (75)	20 (25)	
**Smoking**										
Current smoker	110 (22)	42 (38)	68 (62)	0.922	39 (36)	71 (65)	0.692	82 (75)	28 (25)	0.295
Ex-smoker	125 (25)	45 (36)	80 (64)		51 (41)	74 (59)		89 (71)	36 (29)	
Never smoker	274 (54)	99 (36)	175 (64)		107 (39)	167 (61)		183 (67)	91 (33)	

BMI: body mass index; ^a^ ≤20,000 residents; ^b^ 20,001–500,000 residents; ^c^ >500,000 residents; ^d^ <18.5 kg/m^2^; ^e^ 18.5–24.99 kg/m^2^; ^f^ 25.0–29.9 kg/m^2^; ^g^ ≥30 kg/m^2^.

## Data Availability

Data available on request from the authors.

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
