# Peer review of "Lost in Communication: Do Family Physicians Provide Patients with Information on Preventing Diet-Related Diseases?"

_ijerph, 2022, doi:10.3390/ijerph191710990_

Round 1

Reviewer 1 Report

Dear Authors,

Please find attached the comments regarding the paper. The topic is interesting, however, the paper is missing some information, especially in the methodology section.

Kind regards

Reviewer 2 Report

General comments: The aim of the stud was to assess if patients are provided with recommendations on proper nutrition, physical activity and use of medication from their family physicians. The authors found significant differences between patient declarations of information received concerning physical activity, and diet compared to the recommended use of medication. They concluded that there is room for improvement in providing patients with proper recommendations on diet and physical activity by their family physicians.

Specific comments:

1. Introduction: Although authors provide some rationale for the study, the do not provide a strong justification for it. What are the prominent recommendations for disease prevention? What have previous studies uncovered so far? what are the "numerous and diverse ways to achieve this"? (lines 50-52). What is lacking in those studies that the authors are seeking to answer in this study?

2. Materials and Methods: Can you explain how this sample size was decided upon? You had mentioned that this is a part of a larger study, so can you give a little detail to help readers understand it better? You cited reference #9 as the previous study, but #9 is the W.H.O. Report? Could you recheck your references to ensure that they are corresponding to your in-text citations?

4. Discussion: Line 116 - in-text citation style differs from the numbering the authors have been using throughout manuscript. Is this same as reference #9 in the list of references? Please check.

5. Conclusions: This section can be improved. How may future studies improve upon the limitations of our study?
